# Water–Food Nexus through the Lens of Virtual Water Flows: The Case of India

**Suparana Katyaini** [1,*], **Mimika Mukherjee** [2] **and Anamika Barua** [2]

1   School of Livelihoods and Development, Tata Institute of Social Sciences, Hyderabad 501510, India
2   Department of Humanities and Social Sciences, Indian Institute of Technology Guwahati,
    Assam 781039, India; mimika@iitg.ac.in (M.M.); abarua@iitg.ac.in (A.B.)
*   Correspondence: suparana.katyaini@tiss.edu

**Abstract:** For a water-secure present and future, there is a need for a transition from water scarcity towards water security. This transition necessitates a look at the complex relationships, and interdependencies, between water and other resources, and the institutions governing them. Nexus approach encompasses these interdependencies. This paper focused on the water–food nexus through the lens of the virtual water (VW) flows concept with the aim to explore the role of the VW flows concept in governing the transition towards water security in a water-scarce economy like India. The key findings of the paper suggests that the highest VW outflows are from highly water-scarce states of India, such as Punjab and Andhra Pradesh, and the moderate to highly water-scarce state West Bengal from 1996–2014. Major VW outflows from these states are to other highly water-scarce states, resulting in the concentration of water scarcity. The main priorities for the governance of the water–food nexus in these states emerge from policies and action plans. These priorities are groundwater overexploitation, water and soil pollution, and uncertainty in rainfall and are linked to agricultural intensification. The water footprint-based VW flow analysis has important insights for sustainable intensification of agriculture, and rectification of the unsustainable VW flow patterns. The study concludes that the VW flows concept embodies the water–food nexus and is particularly relevant for the sustainable future of developing and emerging economies, such as India, grappling with water scarcity and challenges of fragmented environmental governance systems.

**Keywords:** India; governance; VW flows; nexus; water footprint; water scarcity; water security





## 1. Introduction

Water plays a central role in ensuring a sense of security and wellbeing. Water resources are finite, unevenly distributed over the landscape, and are prone to misuse and mismanagement [1]. There is continuous pressure on freshwater resources due to increases in population and associated demand, sectoral water requirements, urbanization, changing lifestyles and consumption patterns, and pollution of water sources [2,3].

Due to these multiple drivers of freshwater scarcity, it is increasingly being recognized as a global systemic risk [4,5]. In fact, studies suggest that the freshwater planetary boundary is approaching rapidly [6,7]. Within Sustainable Development Goal 6 on water, a specific target is aimed at addressing water scarcity. Target 6.4 states, "By 2030, substantially increase water-use efficiency across all sectors and ensure sustainable withdrawals and supply of freshwater to address water scarcity and substantially reduce the number of people suffering from water scarcity" [8]. From this target, the urgency to address water scarcity can be inferred for a sustainable present and future. For a water-secure present and future, there is a need for a transition from water scarcity towards water security.

Water security has been defined as, "The capacity of a population to safeguard sustainable access to adequate quantities of and acceptable quality water for sustaining livelihoods, human well-being, and socio-economic development, for ensuring protection against waterborne pollution and water-related disasters, and for preserving ecosystems in a climate

of peace and political stability." [9]. So, a transition from water scarcity to water security necessitates a look at the complex relations and interdependencies between water resources and other resources, along with the institutions and policies that govern them [10].

Transition from water scarcity to water security is also linked to the security of other resources, specifically food security. For instance, Lal (2015) emphasizes the close relationship between food security (availability, access, nutritional quality, retention) and security with respect to water (renewability, availability, quality), soil (quality, resilience), energy (supply, price, dependability), and climate (optimal temperature and moisture regimes, and low frequency of extreme events) [11]. It is also linked to economic (income and access to resources) and political stability (peace and harmony).

More than one in seven people, globally, are food insecure, and almost all of them live in developing countries [12]. To assure food security, the global food production must be increased by 50% by 2030 and 100% by 2050 [13]. This would require an associated increase in the withdrawal of freshwater resources. An important concern is that agroecosystems and related activities are already using 71% of the global freshwater withdrawn [14]. Such concerns can be addressed through an integrative perspective on the governance of natural resources, where the interlinkages between resource use are considered [10]. However, the governance of natural resources, such as water, has long been recognized as complex. The complexity arises from the interconnections within and between natural and social systems [15–17].

The nexus approach is an emerging approach which internalizes the complex interconnections within and between natural and social systems. The nexus approach encompasses the linkages between two or more elements and conceptually links multiple resource use practices (Figure 1). Since 2011, research on the water–food nexus, water–energy nexus, water–food–energy nexus, and water–soil–waste nexus has gained momentum [18]. A core component in the nexus is water. However, an important characteristic of the approach is that all the elements in the nexus are to be viewed with equal importance [19], thereby suggesting that the governance of resources forming the nexus should be polycentric in nature.

Resources, institutions, and security are the three important layers of the nexus approach [20]. In the context of the nexus involving water resources, water security and governing institutions that facilitate the transition towards water security are important considerations. Resource security is an important layer because the nexus approach is intended to reduce the human footprint on planetary boundaries through resource recovery, and increased resource use efficiency. The nexus approach embodies a close relationship between the security of resources like water, soil, and climate, and economic and political aspects [3,11]. The emphasis of this paper is on the water–food nexus, which is especially relevant for the semi-arid and arid areas of the world [21]. The continuous pressure on freshwater resources to support agricultural intensification, uncertainties posed by water-mediated disasters, and land degradation highlight the importance of studying the nexus interlinkages [3,22]. The water–food nexus is viewed as important for advancing food security in a water-scarce world [11,23]. While the nexus approach appears to be promising, the use of analytical methods to systematically evaluate the interlinkages between the resources or inform relevant resource policies has been limited [24]. A systematic review of methods for nexus assessment, carried out by Albrecht, Crootof, and Scott [24] reflects that difference discipline approaches have been used to operationalize nexus assessment. Among the various methodological approaches, footprint-based virtual water (VW) emerged as an important methodological approach from the interdisciplinary field of environmental management.

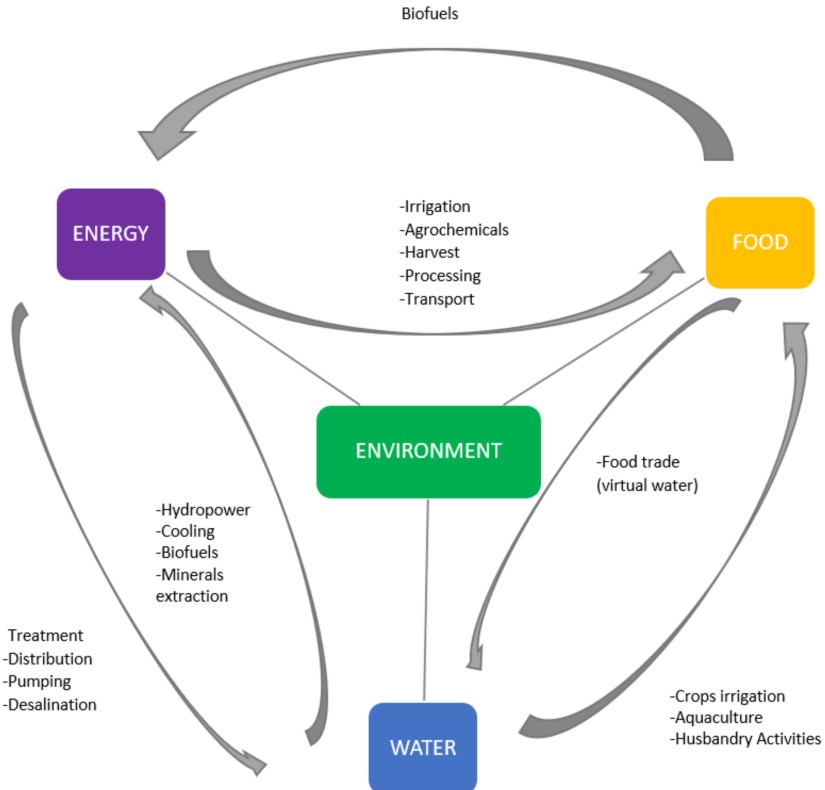

**Figure 1.** Schematic representation of the interdependencies between water, food, energy, and environment. Source: [25].

VW is defined as the water required in the production of a good, and it refers to both the water used and wastewater generated [26] and is linked with the traditional concept of resource usage [27,28]. The idea was first proposed by Prof. Tony Allan to carry out an empirical analysis of water scarcity in the Middle Eastern and North African (MENA) region. It was intended to investigate whether trade flows are aligned with water resource endowments [29]. The idea is central to the discussion on the water–food nexus because it is indicative of water consumption. The VW flows concept has been applied to study water availability, food security, water use efficiency, economic diversification, conflict mitigation, and water scarcity management [29].

With this background, the study aims to explore the question of how the VW flows concept, which is at the heart of the water–food nexus, can play a role in governing the transition towards water security in water-scarce economies. The analysis has been carried out in the context of India. There are two main foci of the analysis: first, to identify the states in India with the highest VW outflows embodied in major agricultural products, such as food grains and oilseeds, and which states these VW outflows go to. Second, there is an analysis of the governance of the water–food nexus in these states with the highest VW outflows. This analysis is intended to identify the priorities for the states to transition from water scarcity to water security. This study is an extension of Katyaini and Barua [30,31] and Katyaini, Barua, and Duarte [32]. The identification of the states with the highest VW outflows has been discussed in Katyaini and Barua [31] and Katyaini, Barua, and Duarte [32]. The two important extensions covered in this paper are the identification of the five main states to which the highest VW outflows go to, embodied in oilseeds. Second, the identification of the key priorities for water–food governance in the states with the highest VW outflows embedded in food grains and oilseeds based on the qualitative analysis of the state-level policies and action plans.

The paper is structured in five sections. Important concepts of the study, such as water scarcity, water security, water–food nexus, and VW flows, are introduced in the first section.

The rationale of considering the case of India is discussed in the second section. The third section is on the materials and methods used in the research. The fourth section is on the results and discussion on the VW flows assessment and the priorities that emerge for the transition towards water security. Key conclusions are presented in the fifth section.

## 2. India: An Important Case to Study the Transition from Water Scarcity to Water Security

There are different analytical approaches to establish the existence and extent of water scarcity [31,33]. India is an important case to study the water–food nexus and how it can aid in the transition from water scarcity to security. Multiple approaches have classified India as a water-scarce economy. For instance, the assessment of annual blue water scarcity, which is a crucial measure, suggests that India (South Asia), and Mexico (Central America) experience moderate to severe water scarcity for a duration of 4-5 months every year, i.e., from February to May or June [34]. Additionally, nearly half the Indian and Chinese population face severe water scarcity for at least 1 month/year [34]. In certain river basins, water scarcity results from a mismatch between freshwater availability and demand. The Ganges basin in India, along with the Limpopo basin in southern Africa and the Murray–Darling basin in Australia, are the river basins where blue water consumption is highest during the period of lowest water availability [35].

There is also growing emphasis on water scarcity at the sub-national scale. In these assessments, highly populous areas and areas with large-scale irrigated agriculture are highly water scarce. Such regions are in India, eastern China, and the Nile delta [36]. The MENA region and parts of South Asia, which include India, are at increased hydrological risks. This is linked to a higher possibility of prolonged droughts coupled with increased variability in rainfall [37].

Furthermore, India is among the water-scarce economies experiencing a highly skewed distribution of water availability in space and time, uncertainties in supplies, and frequent and intense occurrence of hydro-meteorological disasters like floods and droughts. Katyaini and Barua (2016) have drawn on four fundamental approaches, namely, the human water requirement, water resource vulnerability, the incorporation of environmental water needs, lifecycle assessment, and water footprint to map the level of water scarcity in the country [30]. The findings indicate that Punjab, Haryana, Rajasthan, and Gujarat in the north-west, and Andhra Pradesh and Tamil Nadu in the south, experience high water scarcity. There are parts of India which face moderate to high, moderate, and low to moderate water scarcity (Figure 2).

The water withdrawal of the Indian agricultural sector is approximately 85–90%, which is significantly higher than the global average of 70%. Hence, the water–food nexus is a pivotal area of research in the country [3]. In India, 75–80% of the total annual rainfall is concentrated in three months of monsoon season [3]. This leads to a heavy reliance on irrigation to optimize crop production. Irrigation is considered as important for increasing the resilience of the agricultural system, but excessive irrigation can have adverse impacts on the crops, the environment, and water security in India.

In India, states have an important constitutional responsibility regarding the governance of water and agriculture. According to the Constitution of India, the responsibilities of the state and centre are classified into three lists. Water is included in List II i.e., the State List, therefore, water is a state subject. The state carries out the comprehensive multi-sectoral planning, development, and management of the state's water resources, and service deliveries for various water users [38]. However, some decisions are under the purview of the centre and covered in List I, i.e., the Union List. These are the decisions regarding the regulation and development of waters of inter-state rivers, and the inter-state differences and disputes [37]. In India, water governance lags in terms of the pace at which scientific knowledge is incorporated into water policies. Very few states in India govern their water resources through state-specific water policies [30]. Certain reforms are taking place to support decentralized, evidence-based policy decisions. One such important step in the direction of integrated policy management is the formulation of the new draft of

the Science, Technology, and Innovation Policy (STIP) 2020. This draft highlights that water, agriculture, and food security are important thematic areas to address in the future [39,40].

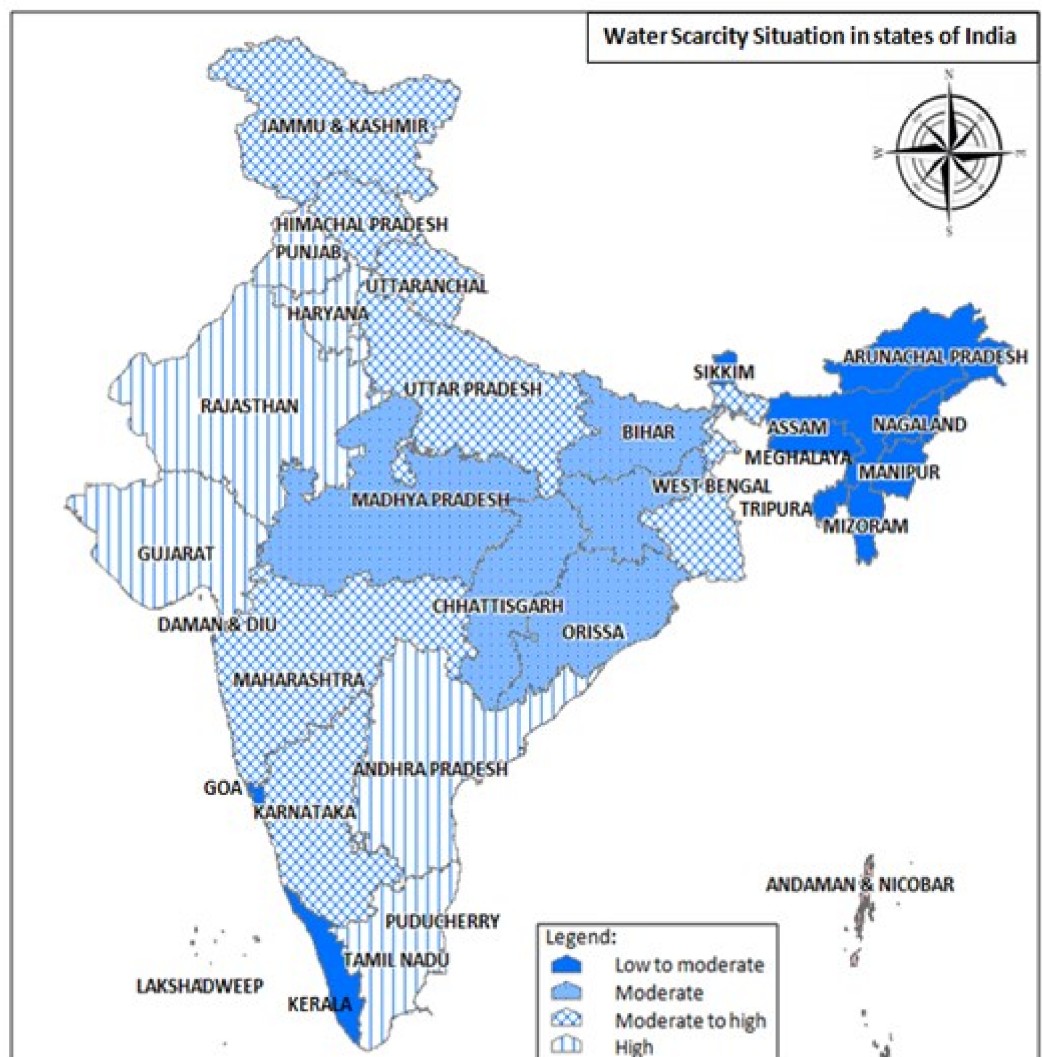

**Figure 2.** Water scarcity situation in states of India. Source: [30].

States have responsibility for many important aspects of agriculture. However, the central government plays a key role in developing national approaches to policy and providing the necessary funds for implementation at the state level. The central government (Union Cabinet) responsibilities include international trade policies and overseeing the implementation of the National Food Security Act (NFSA) of 2013. In 2016, the central government aimed to double farmers' income by 2022–23 through certain important measures such as increasing resource use efficiency, and increasing cropping intensity [41]. The central government's suggested measures act as guidelines for the states. Therefore, state is an important level of governance of the water–food nexus.

The next section emphasizes on the methodology steps undertaken to carry out the study.

## 3. Material and Methods

There are two methodological phases of this study corresponding to the two research objectives (Figure 3).

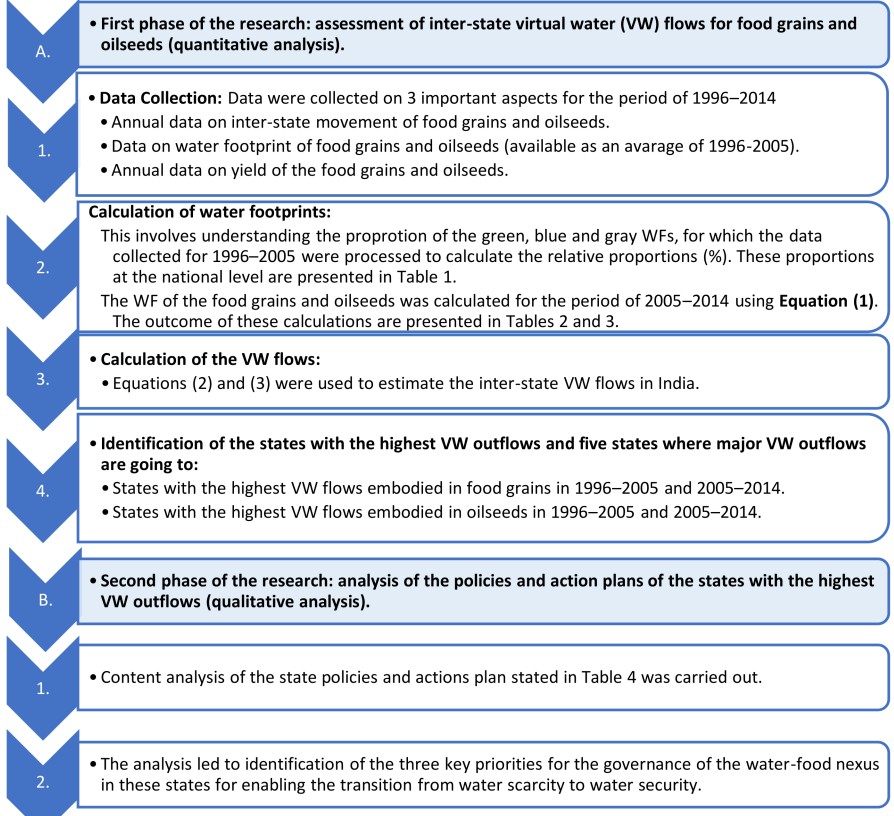

**Figure 3.** Methodological steps.

The first phase involved assessment of the inter-state VW flows embodied in food grains and oilseeds. The food grain categories considered for the analysis were rice, wheat, gram, pulses other than gram, sorghum and pearl millet, maize and millets, and other sorts of grains. These categories were selected because they are viewed as important for achieving food security in India and the inter-state movement of goods data are available for these categories with the Directorate General of Commercial Intelligence and Statistics (DGCIS). Among oilseeds, nine important categories were considered, oilseeds of cotton, oilseeds other than cotton, groundnut oil, mustard oil, castor oil, other vegetable oil, and oilcakes. Appendices A and B refer to the states which are major producers of food grains and oilseeds in India [42]. The data on inter-state movement of these food grains and oilseeds were collected from the annual records of the DGCIS, Ministry of Commerce and Industry for the period of 1996–2014.

The time period of 1996–2014 is crucial for understanding the water–food nexus in India because the period of 1996–1997 to 2005–2006 is classified as the post-reform period, and 2006–2007 to 2009–2010/11 as the period of recovery [43,44]. The significance of these time periods is that impacts of important institutional and technological reforms on water use can be understood through VW flows assessment [31]. For the analysis, we considered 2 time periods of 9 years each, that is, 1996–2005 and 2005–2014. This is because the data on the WF of the food grains and oilseeds are available as the average of 1996–2005 from Mekonnen and Hoekstra [45,46]. Each year is referred to in financial year terms, so the year 1996–97 refers to 1 April 1996–31 March 1997. This format is followed because of the availability of the inter-state movement of food grain and oilseed data.

Water footprint (WF) forms the analytical basis of the VW flows concept. To engage with the discourse on transition from water scarcity to water security, there is a need to consider the differentiated nature of the WF. There are three kinds of WF: blue, green, and gray. Blue water refers to the volume of surface and groundwater consumed in the production of goods and services. The green WF is indicative of the volume of rainwater

consumption in the production process. Gray WF measures the freshwater pollution associated with the production process. In other words, it refers to the volume of freshwater that is required to dilute pollutants to such an extent that the quality of the water remains above agreed water quality standards [47].

A third dataset is needed to estimate the WF for the period of 2005–2014, i.e., the data on yields of the food grains and oilseeds. They were collected from annual records of the Ministry of Agriculture for the period of 1996–2014. Equation (1) was used to estimate the WFs for each of the product (*p*) in a state (*s*) within a time period (*t*) [32].

$$WF_p^s(p,s,t) = WF_p^s(p,s,1996–2005) * \frac{Yield_p^s(p,s,1996–2005)}{Yield_p^s(p,s,t)} \tag{1}$$

The national averages of the proportions of WFs are given in Table 1. The relative proportion of the green, blue, and gray WFs for the 2 time periods 1996–2005 and 2005–2014 are considered to be the same. This is because there was not a significant change in the type of agriculture, i.e., irrigated (blue WF) and rainfed agriculture (green WF).

**Table 1.** National averages of the proportions of water footprints (WFs) in food grains and oilseeds in India.

| Food Grains | Proportion of Different Types of WF (%) | | | Oilseeds | Proportion of Different Types of WF (%) | | |
|---|---|---|---|---|---|---|---|
| | Green | Blue | Gray | | Green | Blue | Gray |
| Rice in the husk | 67 | 22 | 11 | Oilseeds, cotton | 70 | 20 | 10 |
| Rice not in the husk | 67 | 22 | 11 | Oilseeds other than cotton | 92 | 3 | 5 |
| Wheat | 30 | 56 | 14 | Groundnut oil | 84 | 10 | 6 |
| Wheat flour | 30 | 56 | 14 | Mustard oil | 49 | 44 | 7 |
| Gram and gram products | 79 | 2 | 19 | Castor oil | 82 | 15 | 2 |
| Pulses other than gram | 59 | 10 | 31 | Other veg. oil | 93 | 3 | 4 |
| Sorghum and millet | 94 | 2 | 4 | Oilcakes | 89 | 8 | 3 |
| Maize and millets | 91 | 3 | 7 | | | | |
| Other sorts of grains | 81 | 14 | 5 | | | | |

Data sources: [45,46]. The data were processed to estimate percentages of the green, blue and gray WFs.

The total WFs of the food grains and oilseeds for the two time periods 1996–2005 and 2005–2014 are given in Tables 2 and 3. The details of the state-wise WF for 1996–2005 [45,46] and 2005–2014 [48] were used for arriving at the zone-wise WFs. These zone-wise WFs are intended to give an insight into the variations in the water intensity of food grain and oilseed production within India, and the kind of water intensity.

Among the food grain categories, sorghum and pearl millets have the largest WF (5028 m³/ton) in the period of 1996–2005 (Table 2). Approximately 94% of it is green WF, indicating that the major source of water consumed for the production is rainfall (Tables 1 and 2). The smallest WF is for rice in the husk (2070 m³/ton) and wheat (2100 m³/ton). These are the food grains which have the largest blue WF, indicating that they are produced through irrigated agriculture (Tables 1 and 2). In wheat production, the highest proportion of the blue WF is used (56%) as it is an irrigation-intensive crop. Pulses other than gram (31%) have the largest gray WF, suggesting the water pollution is higher in their production (Table 1). Sorghum and pearl millets continue to have the largest WF in 2005–2014, however, it decreased to 3669 m³/ton due to improved yield.

**Table 2.** WFs of food grains (1996–2014).

| Food Grains | Zone-Wise WFs for the 2 Time Periods | | | | | | | | | | | |
| | North-East | | East | | Central | | South | | North | | West | |
| | 1996–2005 | 2005–2014 | 1996–2005 | 2005–2014 | 1996–2005 | 2005–2014 | 1996–2005 | 2005–2014 | 1996–2005 | 2005–2014 | 1996–2005 | 2005–2014 |
|---|---|---|---|---|---|---|---|---|---|---|---|---|
| Rice in the husk | 1844 | 1672 | 1865 | 1501 | 1947 | 1499 | 2295 | 2081 | 2354 | 1931 | 2669 | 2548 |
| Rice not in the husk | 2395 | 2171 | 2421 | 1949 | 2529 | 1947 | 2981 | 2702 | 3057 | 2507 | 3467 | 3309 |
| Wheat | 1805 | 1513 | 1948 | 1851 | 4142 | 3614 | 3800 | 3248 | 1427 | 1256 | 3738 | 2141 |
| Wheat flour | 1827 | 1531 | 1971 | 1873 | 4191 | 3657 | 3681 | 3287 | 1444 | 1271 | 3782 | 2842 |
| Gram and gram products | 2632 | 2445 | 3300 | 2833 | 3561 | 2999 | 4043 | 3680 | 4304 | 4303 | 4698 | 3067 |
| Pulses other than gram | 2285 | 1941 | 3468 | 2995 | 3855 | 3260 | 4511 | 3943 | 1989 | 1787 | 4384 | 3932 |
| Sorghum and pearl millet | 2763 | 1773 | 4323 | 5875 | 4283 | 3512 | 5193 | 4517 | 4796 | 4365 | 5773 | 3147 |
| Maize and millets | 2318 | 2301 | 2595 | 2288 | 2698 | 2464 | 3280 | 2950 | 3226 | 2644 | 3910 | 2197 |
| Other sorts of grains | 2180 | 1937 | 2602 | 2513 | 2581 | 2414 | 2120 | 2114 | 2659 | 2495 | 2874 | 3010 |

The WFs for the period of 2005–2014 are calculated by the authors.

**Table 3.** WFs of oilseeds (1996–2014).

| Oilseeds | Zone-Wise WFs for the 2 Time Periods | | | | | | | | | | | |
| | North-East | | East | | Central | | South | | North | | West | |
| | 1996–2005 | 2005–2014 | 1996–2005 | 2005–2014 | 1996–2005 | 2005–2014 | 1996–2005 | 2005–2014 | 1996–2005 | 2005–2014 | 1996–2005 | 2005–2014 |
|---|---|---|---|---|---|---|---|---|---|---|---|---|
| Oilseeds, cotton | 6135 | 2384 | 7302 | 3831 | 8151 | 6182 | 6871 | 4319 | 7714 | 3867 | 7804 | 4562 |
| Oilseeds other than cotton | 10,746 | 9926 | 13,360 | 12,515 | 14,359 | 11,549 | 17,863 | 15,165 | 2952 | 2331 | 19,183 | 15,868 |
| Groundnut oil | 8290 | 7385 | 9600 | 11,017 | 6409 | 5007 | 11,260 | 10,395 | 8108 | 6837 | 6604 | 5202 |
| Mustard oil | 5502 | 5051 | 6546 | 5267 | 7214 | 5984 | 6018 | 3081 | 5152 | 4487 | 3662 | 2899 |
| Castor oil | 17,859 | 13,559 | 22,034 | 20,880 | 24,116 | 18,886 | 23,696 | 22,138 | 16,844 | 12,383 | 25,803 | 19,579 |
| Other veg. oil | 13,564 | 12,473 | 17,724 | 16,810 | 18,411 | 12,575 | 25,340 | 22,418 | 24,165 | 28,573 | 25,153 | 20,935 |
| Oilcakes | 2316 | 2112 | 2768 | 2595 | 3126 | 2516 | 3638 | 3030 | 3457 | 2968 | 3838 | 3464 |

The WFs for the period of 2005–2014 are calculated by the authors.

Among the oilseeds, castor oil has the largest WF (23,122 m$^3$/ton) in the period of 1996–2005. Approximately 82% of it is green WF. The highest proportion of the green WF is of other vegetable oil (93%) (Tables 1 and 3). The smallest WF is of oilcakes (3344 m$^3$/ton). Except mustard oil, which has the largest blue WF (44%), oilseeds are predominantly produced through rainfed agriculture. The highest proportion of wastewater generation is associated with oilseeds of cotton and is reflected in the gray WF (10%). There was a decrease in the WFs of oilseeds from 1996–2005 to 2005–2014 due to improvement in yield. Other vegetable oil has the largest WF (17,528 m$^3$/ton) from 2005–2014. Castor oil continues to have a large WF, and it has second largest WF from 2005–2014 (16,999 m$^3$/ton). Oilcakes continue to have the smallest WF (2825 m$^3$/ton).

The *VW* outflows embedded in each of the products (*p*) exported from a state (*s*) to other states within a time period (*t*) were calculated as *VWX* (*s*, *t*) using Equation (2). Here, *WF* refers to the total *WF* of a product, and *Q* refers to the quantity of the product.

$$VWX(s,t) = \sum_p WF_p^s(s,p,t) * Q_p^s(s,p,t) \tag{2}$$

A similar expression is used for calculating the virtual water imports (*VWMs*) to indicate the *VW* inflows. The difference between the *VWM* and *VWX* for each of the states reflects the actual pressure on scarce freshwater resources of the producing state.

The final important methodological step in the inter-state VW flows assessment is the calculation of the "theoretical water balance". It is a measure of potential pressure on freshwater resources of the importing state that is averted by imports of the products from other states. It is measured for each of the states through Equation (3). Here, *Q* refers to the quantity of the product, $s_e$ is the state the *VW* outflows are from, and $s_i$ is the state where the *VW* outflows are going to.

$$VWBT_{(p,s,t)} = Q_{(s_e,s_i,p)} * (WF_{(s_i,p)} - WF_{s_e,p}) \tag{3}$$

For the detailed intermediate methodological steps to estimate the VW flows at the state level, please see Katyaini, Barua, and Duarte [32].

The second methodological phase was to carry out a critical analysis of the policies and action plans of the states with the highest VW outflows, as these states are a concern from the water-scarcity perspective. The analysis was aimed at identifying the key priorities for the governance of the water–food nexus in these states for enabling the transition from water scarcity to water security. The important policy and action plan documents available for different states of India for carrying out this analysis are listed in Table 4. These are state water policies, state water missions, and state agriculture missions as a part of the state action plans on climate change (SAPCC), and other relevant publications on the governance of water and agriculture in the public domain. An important observation in this phase was that different degrees of detail are available for states in terms of the policies, action plans, and acts. As a result, we have tried to assimilate the information relevant to understanding the priorities of the states emerging from these documents. The key findings on the two foci of the study are discussed in the Results and Discussion section.

**Table 4.** Important policy and action plan documents available for different states of India.

| Document | Brief Description |
|---|---|
| State water policies | Three versions of national water policy exist in India: 1987, 2002, and 2012. Around 14 of the Indian states have formulated and implemented a state-specific water policy to govern and manage their water resources. These refer to different years and are based on the national water policy. These states are Tamil Nadu, Andhra Pradesh, Rajasthan, Himachal Pradesh, Karnataka, Maharashtra, Uttar Pradesh, Jharkhand, Orissa, Goa, Kerala, Rajasthan, Chhattisgarh, and Madhya Pradesh. Some states like Punjab only have a draft of the state water policy [31,48]. |
| State action plan on climate change (SAPCC) 1. State Water Mission 2. Mission for Sustainable Agriculture/State Agriculture Mission | The Ministry of Environment, Forest and Climate Change motivated the state governments to prepare SAPCC aligned with the strategies outlined in the National Action Plan on Climate Change (NAPCC). The NAPCC came into existence in 2008. Following the guidelines, 32 states/union territories have formulated SAPCC in subsequent years. These are Andaman and Nicobar Islands, Andhra Pradesh, Telangana, Arunachal Pradesh, Assam, Bihar, Chandigarh, Chhattisgarh, Gujarat, Haryana, Himachal Pradesh, Jammu and Kashmir, Jharkhand, Kerala, Karnataka, Lakshadweep, Madhya Pradesh, Maharashtra, Manipur, Meghalaya, Mizoram, Nagaland, Odisha, Puducherry, Punjab, Rajasthan, Sikkim, Tamil Nadu, Tripura, Uttarakhand, Uttar Pradesh, and West Bengal [49]. In the action plans, water and agriculture are two of the eight important themes. The water mission aims to ensure integrated water resource management to conserve water, minimize wastage, improve water use efficiency, and ensure more equitable distribution. The agriculture mission aims at making agriculture more productive, sustainable, and climate resilient. Water management for agriculture is also an element of it [49]. Since our focus is the water–food nexus and its governance, we referred to the State Water Missions and State Agriculture Missions. |
| Other state-specific policy, plans, and acts on water and agriculture | The state departments have also formulated other state-specific policy, plans, and acts on water and agriculture. These are for very specific purposes like subsoil water conservation, groundwater, etc. [48]. |

## 4. Results and Discussion

From the inter-state VW flows analysis for all the states of India, it was found that the state of Punjab from the northern zone has the highest net VW outflows embodied in food grains in both time periods of 1996–2005 and 2005–2014. For oilseeds, Andhra Pradesh from the southern zone and West Bengal from the eastern zone emerged as the states with the highest net VW outflows from 1996–2005 and 2005–14, respectively. Another important finding is that the quantum of VW outflows embodied in food grains is much larger than the oilseeds. The focus of the results and discussion hereafter is only on the three states with the highest VW outflows. For details on the findings of VW flows assessment at the national level, zonal level, and for other states, please see Katyaini and Barua [31] and Katyaini, Barua and Duarte [32]. The results and discussion on each of the states are structured into four parts: brief description of the state, analytical findings on the net VW

outflows, three key concerns identified from the analysis of the state policy and action plan, and the key measures planned to address these concerns in these documents.

*4.1. Punjab*

Punjab has been the focus of food grain production since the Green Revolution started in 1960s. This was enabled by cultivation in 83% of its geographic area, which is significantly higher than the national average. Agriculture also employs approximately 70% of the population [50].

From 1996–2005, Punjab had the highest net VW outflows, i.e., −4.589 TL/year for food grains (Figure 4). These VW flows from highly water-scarce Punjab are to other highly water-scarce states (Gujarat and Andhra Pradesh), moderate to highly water-scarce states (Maharashtra and Karnataka), and the moderately water-scarce state of Madhya Pradesh. From 2005–2014, Punjab continued to have the highest VW outflows. The outflows increased to −5.928 PL/year, which is equivalent to −5928 TL/year [31]. The VW flows from Punjab to Maharashtra, Gujarat, and Karnataka continued from 2005–2014, with the addition of Tamil Nadu and Assam as the two other states (Figure 5). Tamil Nadu is highly water scarce like Punjab, and Assam has is low to moderate water scarcity. It can be inferred from these VW outflows of Punjab that water scarcity is not being distributed throughout the states.

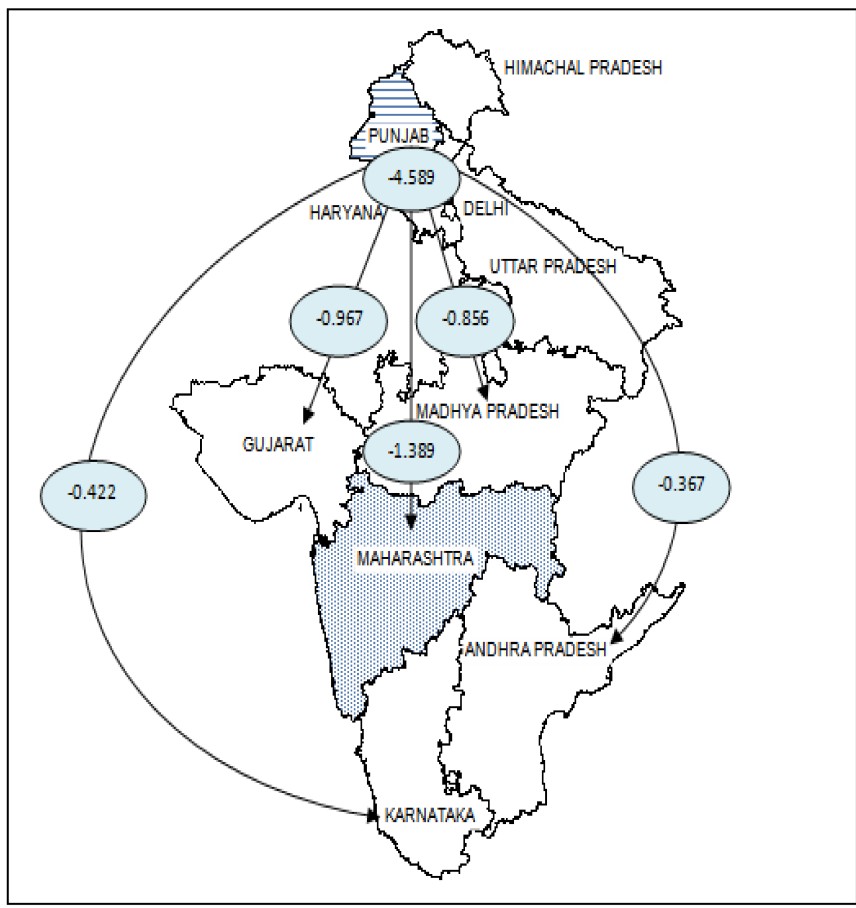

**Figure 4.** Five major VW outflows from Punjab, the state with highest water losses from 1996–2005 (in TL/year).

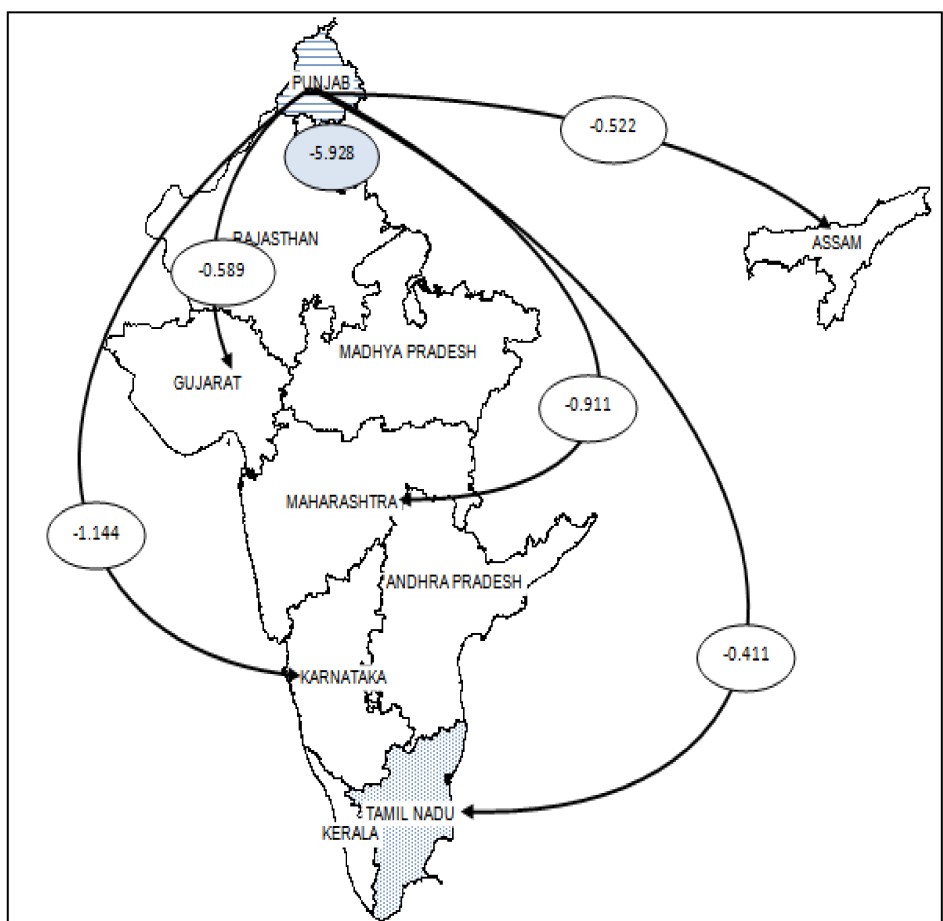

**Figure 5.** Five major VW outflows from Punjab, which had the highest water losses from 2005–2014 (in PL/year).

From the analysis, the three key concerns that emerged for governance of the water–food nexus in Punjab are intensive rice–wheat cropping systems (associated with government subsidy), overexploitation of groundwater resources, and water pollution due to high cropping intensity.

Although Punjab's agro-climatic zones are suitable for the production of maize, rice, jowar (sorghum), bajra (millets), and cotton in the Kharif (autumn) season, and wheat, gram, barley, and rapeseed in the *Rabi* (spring) season, there is dominance of the intensive rice–wheat system (Figure 6) [51,52]. The intensive rice–wheat system has led to Punjab being the second highest producer of wheat and the third highest producer of rice, with contributions of 11.36% and 18.41% to the total rice and wheat production in India [51].

The intensive rice–wheat cropping system is supported through subsidies provided on water, electricity, and fertilizers to farmers. Most of the farmers (63%) have small landholdings of less than 4 hectares, therefore, they rely heavily on these subsidies to maximize their profits. For intensified production, high yielding varieties (HYVs) of wheat and rice were encouraged [53]. Since 2000–2001, the combined area under rice and wheat cultivation has increased to 75% of the total cropped area in Punjab.

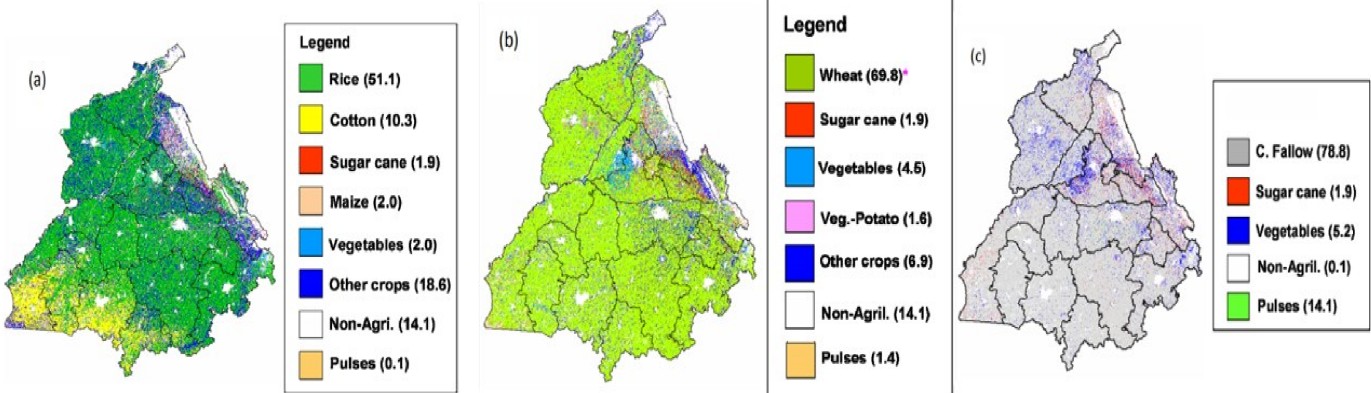

**Figure 6.** Cropping pattern of Punjab in (**a**) Kharif, (**b**) Rabi, and (**c**) summer. Values in brackets represent the percentage of area under each crop with respect to net sown area. Source: [49].

The second concern is the overexploitation of groundwater resources, which is linked to intensive production. Intensification has led to 98% of the cultivable area coming under groundwater-dependent assured irrigation. In the VW flows analysis, groundwater usage in irrigation is reflected as a part of the blue WF. The overexploitation of groundwater was enabled by free or subsidized power for farmers. Unsustainable extraction of groundwater is evident in the sharp increase in the number of tube wells, i.e., from 0.19 million from 1970–1971 to 2.3 million in 2010. This was associated with a sharp decline in the groundwater table from 42 cm (1997–2002) [54] to 75 cm (2002–2006) [55]. The continued overexploitation of groundwater is evident in 109 of 138 groundwater assessment units (Figure 7).

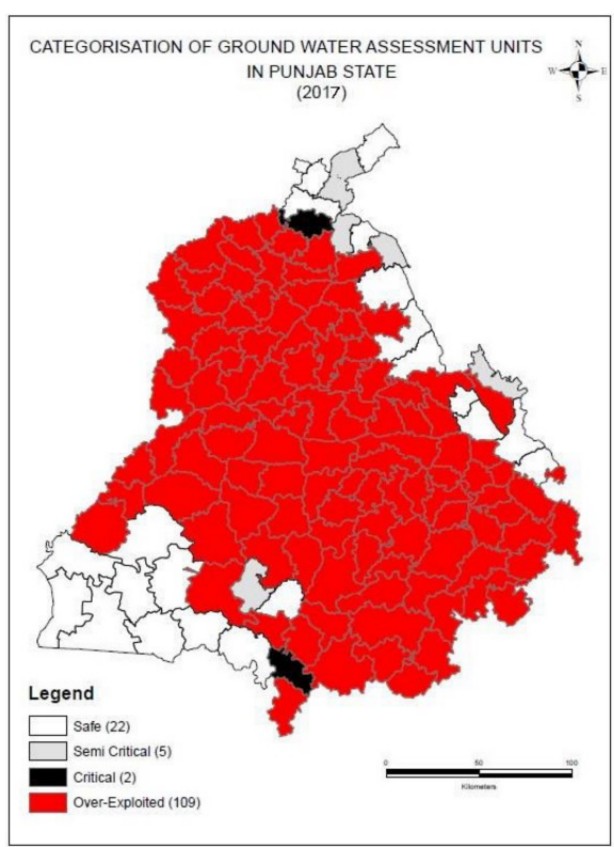

**Figure 7.** Categorization of groundwater assessment units in Punjab state (2017). Source: [56].

The third concern is excessive use of nitrogenous fertilizers and pesticides, resulting in residual toxicity of soil and water. It is associated with a high cropping intensity (189%) [50]. For instance, the Punjab fertilizer usage of 237.05 kg/ha from 2009–2010 was much higher than the national average of 135.27 kg/ha [57]. This led to an imbalance in soil micronutrient composition. Similarly, the pesticide consumption in Punjab (923 g/ha) [58] was more than double the national average consumption (381 g/ha) [57]. In the VW flows analysis, the excessive use of nitrogenous fertilizers and pesticides is reflected in the gray WF.

To address these concerns, some of the state-specific measures have been adopted. For dominance of the intensive rice–wheat system, crop diversification has been suggested. This would also lead to a change in the VW outflows from the state because of the differences in the WFs of producing different crops (Tables 2 and 3). To prevent overexploitation of groundwater resources, the Punjab Preservation of Subsoil Water Act, 2009 encourages planting of paddies in sync with the onset of monsoons. It prohibits farmers from sowing paddy nurseries before 10 May and the transplantation of paddies before 10 June [50]. A strategy in this regard is also proposed in the SAPCC (2014). This is to use aquifers as a water source during dry periods, and a storage reservoir during wet periods. Further, regular monitoring and restriction of groundwater withdrawal whenever and wherever required would discourage overexploitation of groundwater. Improved water use efficiency may also reduce the unsustainable withdrawal of groundwater resources. Increasing water use efficiency by 20% is a state target that can be achieved through enhancement of wastewater reuse, micro-irrigation, and avoiding leakages in the water distribution system.

The third concern on curbing water pollution due to excessive use of fertilizers has received very little attention. The Punjab SAPCC (2014) states that the use of herbal pesticides and organic fertilizers is an important measure for reducing fertilizer and pesticide pollution.

From the institutional mapping on governance of water resources and agriculture in Punjab's SAPCC (2014) (Appendices C and D), the common gap of fragmentation in environmental resource governance can be seen. It is evident in the separate agencies and departments involved in the management and distribution of surface and groundwater resources. However, some indications of integrating the water–food nexus can be gauged from the involvement of the Department of Irrigation and Department of Agriculture in the management of surface and groundwater resources and the involvement of the Department of Soil and Water Conservation in agricultural governance and management.

### 4.2. Andhra Pradesh

Oilseeds form an important part of Andhra Pradesh's agricultural production. Around 37% of the state's geographic area is under cultivation, and 64% of it is rainfed. The annual rainfall varies from 70–150 cm in the coastal areas and 30–50 cm in the driest region. Agriculture engages more than 60% of the state's population [59].

From 1996–2005, highest VW outflows for oilseeds within India were from Andhra Pradesh (0.265 TL/year). A significantly large proportion of these VW outflows were for cotton and groundnut, as these are the two major crops produced in the state (Figure 8). The VW outflows of Andhra Pradesh are from a highly water-scarce state to the other highly water-scarce states of Haryana and Delhi, moderate to highly water-scarce Maharashtra, and moderately water-scarce Madhya Pradesh and Chhattisgarh (Figures 2 and 8).

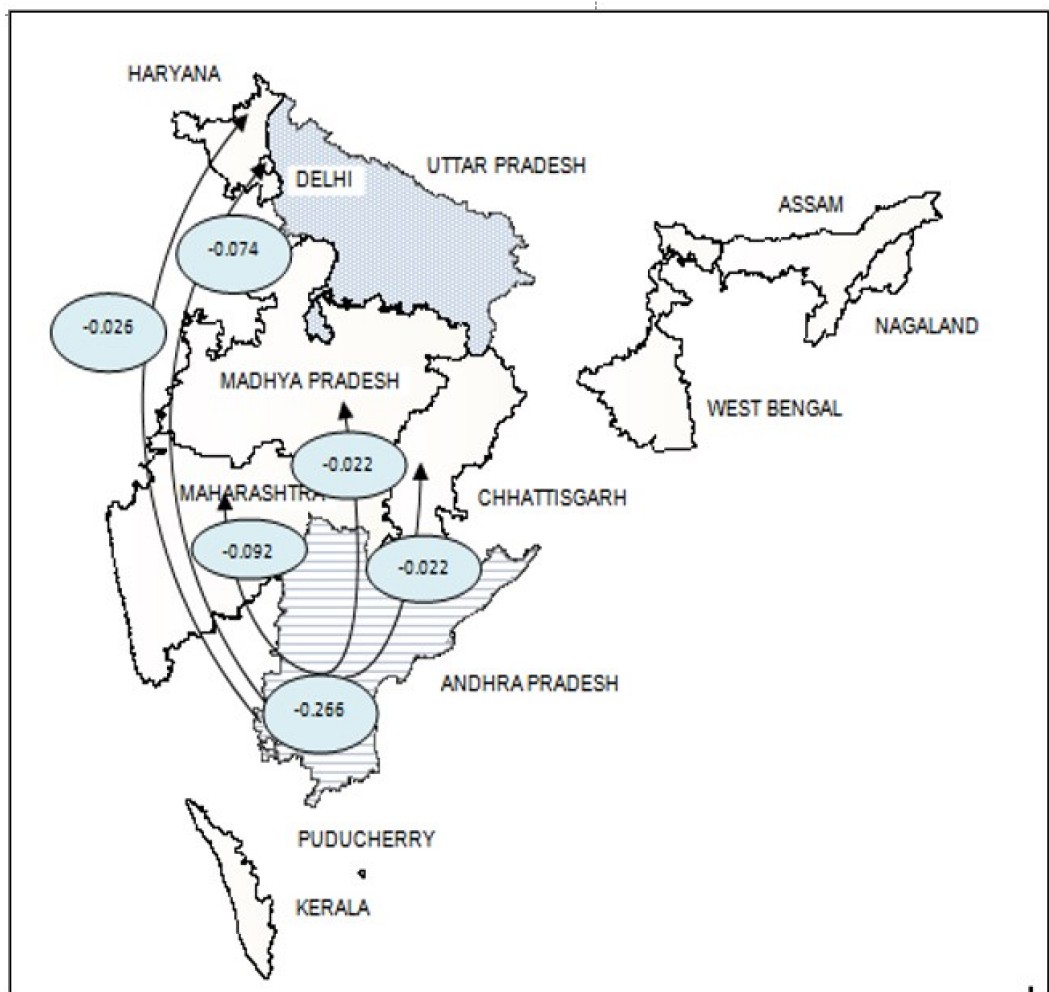

**Figure 8.** Five major VW outflows from Andhra Pradesh, which had the highest water losses from 1996–2005 (in TL/year).

Three major concerns for water–food nexus governance have emerged from our analysis of Andhra Pradesh state water policy (2008), state water mission, and state agriculture mission (2012). These are high dependence on rainfed agriculture, high exposure to water-mediated disasters, and high cropping intensity.

High dependence on rainfed agriculture (60% of the total) makes the state vulnerable to erratic rainfall patterns. High exposure to water-mediated disasters like cyclones and floods along the large coastline of the state affects the output of *Rabi* crops. For instance, a decline of 23.61% was reported from 2008–2009 to 2009–2010 [59]. Andhra Pradesh also has a long history of droughts with the occurrence of droughts twice every five years [60]. High vulnerability to droughts has led to high dependence on tube wells for irrigation, through which groundwater is drawn. From 2009–2010, groundwater dependence accounted for around 42.3% of the irrigation [59].

The cropping intensity was 126% from 2009–2010 in Andhra Pradesh [59]. The high cropping intensity has led to soil nutrient deficiency and, specifically, deficiencies of macro-, micro-, and secondary nutrients have been reported in the rainfed areas [59].

Several measures are proposed to address these concerns in the state water policy and the state water mission, and the state agriculture mission emphasized the need for the water–food nexus approach to transition from water scarcity to security. With particular reference to coping with droughts, water harvesting is proposed and increases in water use efficiency in the Andhra Pradesh SAPCC (2012). Measures such as the promotion of less water-intensive crops and full utilization of surface water irrigation potential and groundwater potential to stabilize production are suggested [37]. Specific to groundwater,

strict regulation of groundwater extraction and recharging of aquifers are proposed. To address the soil nutrient deficiency, the replacement of inorganic fertilizers with bio-fertilizers, reducing the use of synthetic pesticides, and implementation of integrated nutrient management systems for balanced nutrition of crops are proposed. These measures would reduce the wastewater generation in oilseed production, which would be reflected as a decrease in the gray WF and a subsequent decrease in the VW content.

*4.3. West Bengal*

Oilseeds form an important part of agricultural production in West Bengal. Approximately 65.25% of the geographic area of West Bengal is cultivable land, and 46% of it is under rainfed agriculture [61]. West Bengal receives approximately 175 cm of annual rainfall which has an influence on rainfed agriculture [62]. Around 64% of the state population is engaged in agriculture [61].

In the period of 2005–2014, West Bengal emerged as the state with the highest net VW outflows (6.788 TL/year) for oilseeds (Figure 9). Mustard and groundnut form a considerably large part of these VW flows as these are the main oilseeds produced in the state. The major VW outflows from West Bengal are to the other moderate to highly water-scarce states of Uttar Pradesh and Maharashtra, moderately water-scarce Bihar and Madhya Pradesh, and low to moderately water-scarce Nagaland (Figures 2 and 9).

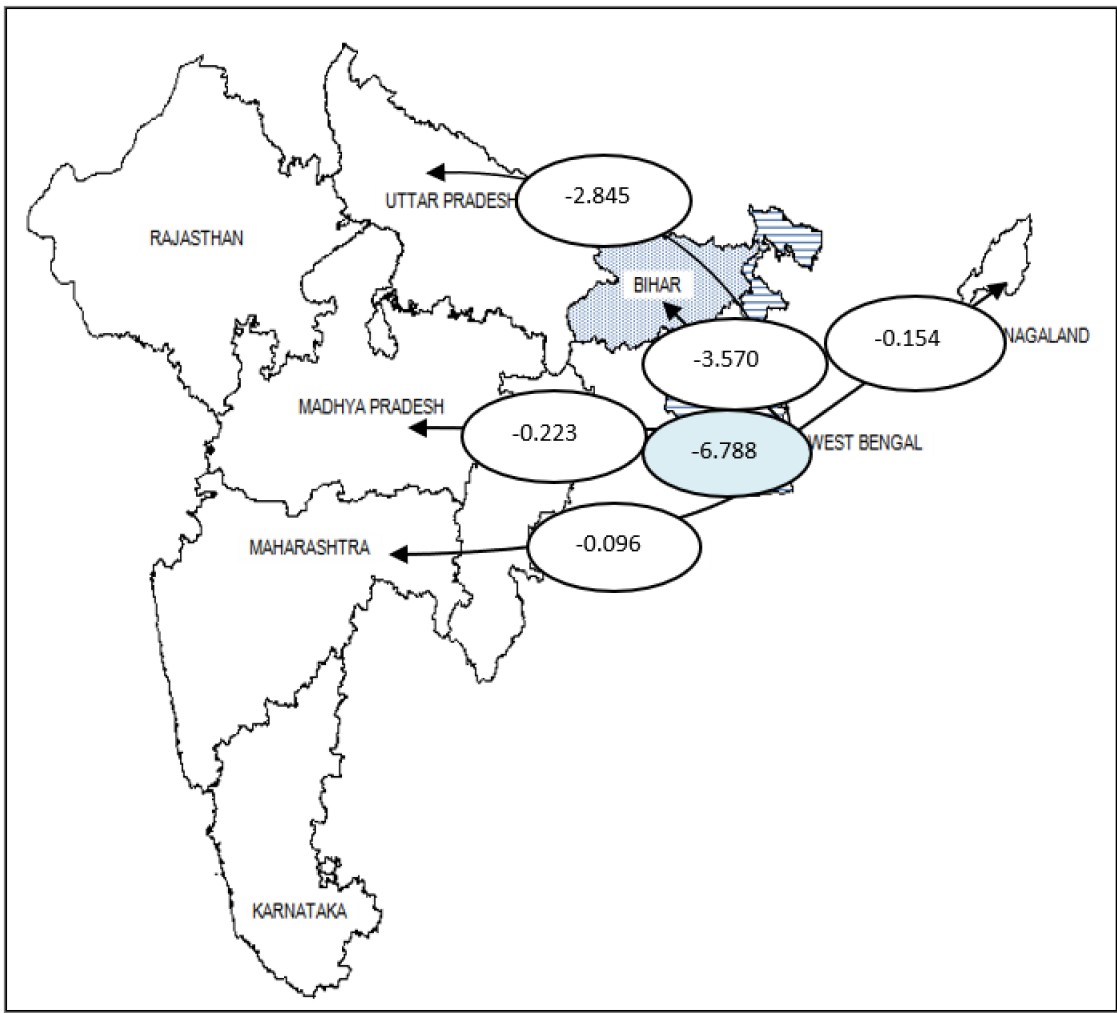

**Figure 9.** Five major VW outflows from West Bengal which had the highest water losses from 2005–2014 (in TL/year).



The three key concerns identified through our analysis for the governance of the water–food nexus in West Bengal are the extreme rainfall events, groundwater overexploitation and quality issues, and high cropping intensities, which stress both soil and water resources.

Regarding the first concern, 77% of the rain is experienced during the monsoon months of June to September. This is associated with 42% of the geographic area being prone to floods, and water logging in the areas where rainfed agriculture is practiced [61].

Minor irrigation plays a key role in the irrigated agriculture, as major and medium irrigation schemes cover only 2.44% of the total irrigated area. Minor irrigation schemes are largely based on groundwater. Groundwater overexploitation for agriculture has led to the critical and semi-critical status of many blocks (Figure 10). The quality of groundwater resources is becoming a concern in the state because of issues of high salinity, and high concentrations of arsenic and fluoride have been reported in 140 of the 341 blocks [61].

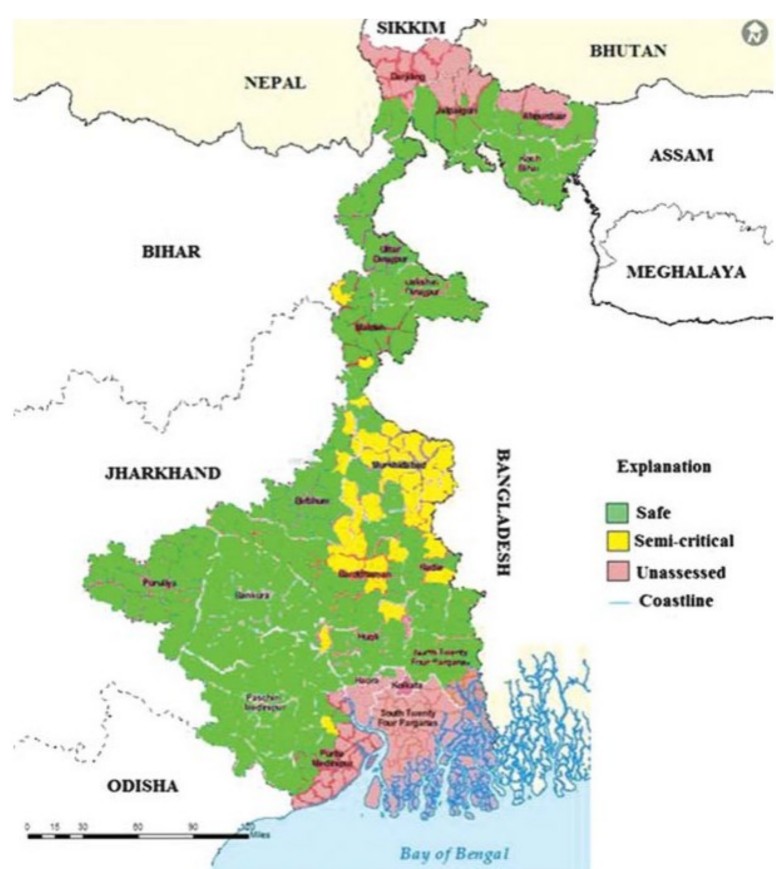

**Figure 10.** Critical, semi-critical, and safe groundwater blocks of West Bengal. Source: [62,63].

The third concern is high cropping intensity (185%) in West Bengal because it is among the highest in India. Bengal's oilseed production witnessed a significant increment to 0.55 million tons from 0.24 million tons in the last 10 years. High cropping intensity exerts immense pressure on the soil and water resources. In fact, four of the six agro-climatic zones have been identified as stressed zones. Degradation of land has occurred, and the soil often requires external intervention to improve its condition [61].

Certain measures have been suggested to address these concerns. For instance, addressing groundwater overextraction and poor quality is being prioritized through the West Bengal Ground Water Resources Management, Control and Regulation Act (2005) [64]. Replenishment of the water tables by efficient allocation and use of the available resources has been suggested. To prevent soil erosion and degradation of soil quality, green manures and sequential cropping methods are promoted in the state. An integrated farming system (IFS) is also being considered to improve farm productivity.

## 5. Conclusions and Way Forward

The WF-based VW flow analysis epitomizes the water–food nexus and is significant in identifying the unsustainable VW outflows from water-scarce areas. Such an analysis can also aid in identifying priorities for the governance of the water–food nexus. Three key conclusions emerge from our study. The first conclusion is that in the highly water-scarce states, planning and implementation of sustainable intensification of agriculture are crucial for achieving water and food security. This finding is in agreement with Pretty et al.'s [65] findings from Africa. In the context of India, sustainable intensification of food grains and oilseeds needs to be planned differently as the nature of the water intensity of their production is different. Further, oilseeds have received significantly less policy attention in India in comparison to food grains. As the policy focus on oilseeds is growing, sustainable intensification of oilseeds becomes important. This is where the well-informed governance of the water–food nexus can play a key role.

The second key conclusion is that the pressure on the freshwater resources of the highly water-scarce states can be reduced by diversifying the production areas. This can be achieved by using the VW flows analysis to support interventions in area with low to moderate water-scarcity which are agro-climatically suitable for the production of specific categories of food grains and oilseeds (Appendix A). This would reduce the concentration of water scarcity and may lead to the distribution of water scarcity. These measures call for joint decision making at multiple levels of governance and the involvement of crucial stakeholders such as farmers, civil society organizations, and concerned government departments for water, agriculture, and food security. Since agricultural water needs and crop yields are specific to the hydrological and geographical features of an area, the state-specific plans need to consider these features in congruence with the local production capabilities. In this process, the states can learn from each other about the possible pressures on the freshwater resources due to certain agricultural production decisions.

The third key conclusion is that although linkages within the water–food nexus are understood to some extent, there is a need for deeper policy engagement with it. A deeper policy engagement would be particularly relevant for the sustainable future of developing and emerging economies grappling with the challenges of water scarcity and fragmented environmental governance systems. This requires further integration of linkages to other resources, such as soil and energy through socio-ecological–economic systems research. The inclusion of energy with the water–food nexus is important for a comprehensive plan to overcome the current and emerging challenges [11]. One of the important emerging challenges for transformation towards water and food security is climate change [4,66]. Climate change impacts water availability and, thus, has both direct and indirect impacts on the agricultural sector. These impacts vary with the location and adaptive capacity to climate risks [67]. Specifically, in South Asia, a decline in the yield of major crops has been associated with climate change [68]. This adds to the structural economic problems, governance challenges, and ecosystem threats [69]. Understanding and mapping the vulnerabilities associated with climate change through innovative research design paves the way for future research on resource conservation and sustainable utilization.

**Author Contributions:** S.K. was engaged in all the stages of developing the research article from conceptualization to writing of the article. S.K. led the data curation, formal analysis, and writing the original draft, and reviewing and editing. M.M. supported in data curation, formal analysis and writing of the article. A.B. provided important direction in all stages of the research article with emphasis on the conceptualization, methodology, formal analysis and writing. All authors have read and agreed to the published version of the manuscript.

**Funding:** We would like to acknowledge the doctoral research fellowship grant 11614104 provided by the Indian Institute of Technology Guwahati, India to carry out this study.

**Institutional Review Board Statement:** Not Applicable.

**Informed Consent Statement:** Not Applicable.

**Data Availability Statement:** The data supporting the reported results can be found in the doctoral thesis submitted to Indian Institute of Technology Guwahati, Assam, India.

**Acknowledgments:** We would like to thank the anonymous reviewers for their valuable suggestions which were significant in improving the discussion in the paper.

**Conflicts of Interest:** The authors declare no conflict of interest.

## Appendix A

**Table A1.** Major Food Grain-Producing States (Zone-Wise).

| Food Grains | Major Food Grain-Producing States Zone-Wise | | | | | |
|---|---|---|---|---|---|---|
| | North | North-East | East | Central | West | South |
| Rice | Haryana, Punjab, Uttar Pradesh | Assam | Bihar, Orissa, West Bengal | Chhattisgarh | | Andhra Pradesh, Karnataka, Tamil Nadu |
| Wheat | Haryana, Punjab, Uttar Pradesh, Uttarakhand | | Bihar, West Bengal | Madhya Pradesh | Gujarat, Maharashtra, Rajasthan | |
| Sorghum and Pearl millet | Haryana, Jammu and Kashmir, Uttar Pradesh | | | Chhattisgarh, Madhya Pradesh | Gujarat, Maharashtra, Rajasthan | Andhra Pradesh, Karnataka, Tamil Nadu |
| Maize and millet | Himachal Pradesh, Uttar Pradesh, Uttarakhand | Arunachal Pradesh | Bihar, Jharkhand | Chhattisgarh, Madhya Pradesh | Gujarat, Maharashtra, Rajasthan | Andhra Pradesh, Karnataka, Tamil Nadu |
| Gram | Haryana, Uttar Pradesh | | Bihar, Jharkhand | Chhattisgarh, Madhya Pradesh | Gujarat, Maharashtra, Rajasthan | Andhra Pradesh, Karnataka |
| Pulses other than gram | Uttar Pradesh | | Bihar, Jharkhand, Orissa | Chhattisgarh, Madhya Pradesh | Gujarat, Maharashtra, Rajasthan | Andhra Pradesh, Karnataka, Tamil Nadu |
| Other sorts of grains (barley) | Haryana, Himachal Pradesh, Jammu and Kashmir, Punjab, Uttar Pradesh, Uttarakhand | | Bihar, Jharkhand, West Bengal | Chhattisgarh, Madhya Pradesh | Rajasthan | Tamil Nadu |

## Appendix B

**Table A2.** Major Oilseed-Producing States (Zone-Wise).

| Oilseeds/oils | Major Oilseed-Producing States Zone-Wise | | | | | |
|---|---|---|---|---|---|---|
| | North | North-East | East | Central | West | South |
| Oilseeds, cotton | Punjab, Haryana | | | Madhya Pradesh | Rajasthan, Gujarat, Maharashtra | Andhra Pradesh, Karnataka, Tamil Nadu |
| Oilseeds other than cotton (soyabean, groundnut, rape, and mustard) | | Soyabean production since ancient times | | Madhya Pradesh, Chhattisgarh | Maharashtra, Rajasthan, Gujarat | Andhra Pradesh, Karnataka, Tamil Nadu |
| Groundnut oil | | | | | Gujarat, Maharashtra, Rajasthan | Andhra Pradesh, Karnataka, Tamil Nadu |
| Mustard oil | Haryana, Uttar Pradesh | Assam | West Bengal | Madhya Pradesh | Rajasthan, Gujarat | |
| Castor oil | | | Orissa | | Rajasthan, Gujarat | Andhra Pradesh, Karnataka, Tamil Nadu |
| Other veg oil (soyabean, sunflower, sesame) | Haryana, Punjab, Uttar Pradesh | Assam | Orissa, West Bengal | Madhya Pradesh, Chhattisgarh | Gujarat, Maharashtra, Rajasthan | Andhra Pradesh, Karnataka, Tamil Nadu |
| Oilcakes (soyabean, groundnut, rape, and mustard) | | Soyabean production since ancient times | | Madhya Pradesh, Chhattisgarh | Maharashtra, Rajasthan, Gujarat | Andhra Pradesh, Karnataka, Tamil Nadu |

## Appendix C

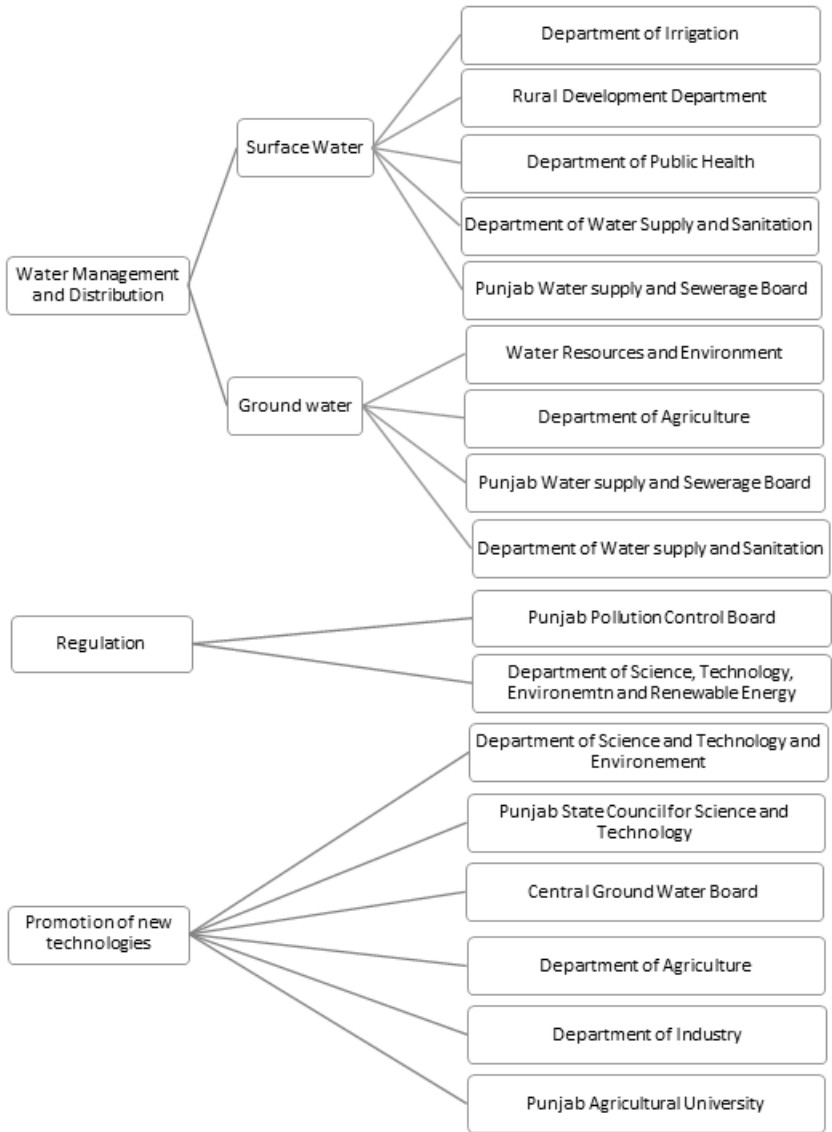

**Figure A1.** Institutions Managing Water in Punjab. Source: [48].

**Appendix D**

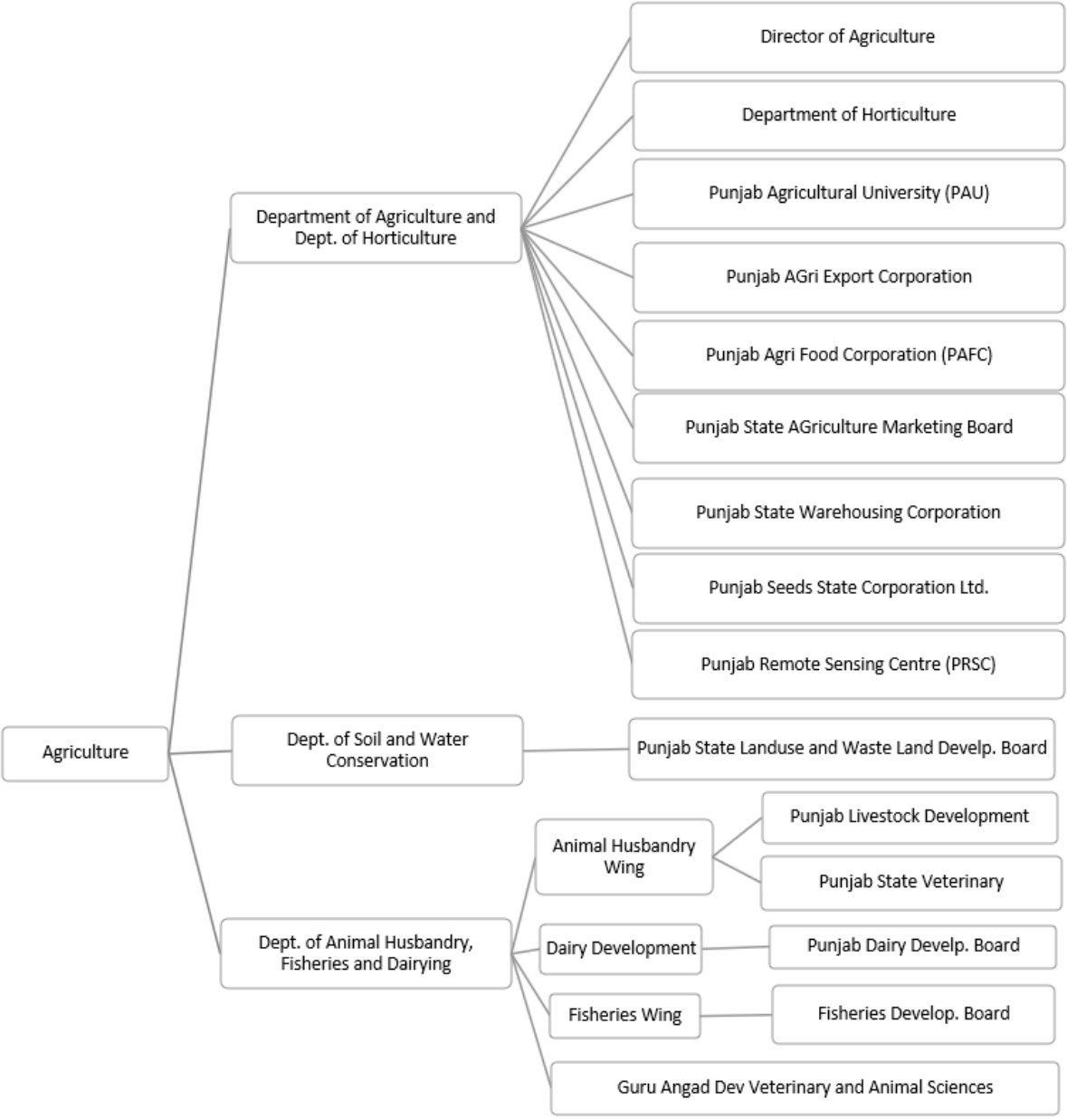

**Figure A2.** Institutions Managing Agriculture in Punjab. Source: [48].

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
