# Peer review of "Water–Food Nexus through the Lens of Virtual Water Flows: The Case of India"

_water, doi:10.3390/w13060768_

Round 1
Reviewer 1 Report
Work carried out in this paper is very novel. Its scientific and societal importance is very high. However, the methodology section is very poorly written to understand the logic and follow the result and discussion.
-All the figures are of poor resolution and hence hard to read any information. Please consider replacing them with high resolution.
- Conclusion and abstract sections need to be fine tuned to summarize the key findings of this study and way forward.
- Please find the attached reviewed paper for more comments.

Reviewer 2 Report
The manuscript “Water-Food Nexus through the lens of virtual water flows: A case of India” is an interesting interdisciplinary study challenging the necessity transition from water scarcity towards water security. The water quality and quantity is the challenge of 21st century that we can’t overlooked specially at contemporary climate changes.
The water-energy-food nexus is rapidly expanding in scholarly literature and policy settings as a novel way to address complex resource and development challenges. The nexus approach aims to identify tradeoffs and synergies of water, energy, and food systems, internalize social and environmental impacts, and guide development of cross-sectoral policies. However, while the WEF nexus offers a promising conceptual approach, the use of WEF nexus methods to systematically evaluate water, energy, and food interlinkages or support development of socially and politically-relevant resource policies has been limited.
The nexus concept can be operationalized as an analytical tool by utilizing approaches that address the four key features —innovation, social and political context, collaboration, and implementation in policy and practice. Promising approaches we identified include using interdisciplinary and mixed-methods, and incorporating transdisciplinary or participatory approaches. Further, analyses target policy- and community-relevant scales. Interdisciplinary and mixed-method approaches that combine quantitative and qualitative methods from multiple disciplines are needed to address the physical and social aspects of water, energy, and food systems. The contribution of social science approaches here is significant, particularly for understanding the social and political context of WEF interactions and feedbacks for resources efficiency, policy integration, and sustainable development. While data from multiple sectors are often utilized, transdisciplinary and participatory approaches that work with stakeholders, decision-makers, and policy-makers in water, energy, and food fields can help align nexus research with policy needs and support its utilization in practice. Further, nexus methods and tools must be made available for use by practitioners and researchers alike.
This is a well conducted study, some of the conclusions/ideas presented here could be applicable not only in Indian circumstances.
The abstract and introduction part is well written and contains good literature review.
The part Material and Methods – could be little bit more reader friendly. Starting by defining the used methods and its importance… and continue by explaining the findings. I suggest to include flow chart for the methodology – it helps the reader not to get lost in this part.
The part of Results is good one. It contains enough information for reader to read the conclusion chapter but I recommend to split parts Results and Discussion. All the findings confirm importance of this study. It is very interesting for reader, or researcher in this field.
To conclude, I recommend authors to reorganize manuscript a bit (Conclusion, Discussion) and make it shorter and add /check the references (web of science (science direct) or Scopus...or other databases) to proof its scientific soundness.
Also I have some formal notes for corrections:
- at the chapter titles are not necessary used the colon: e.g. (line 469) 5. Conclusion and way foreward:
- Figure Nr. 9 could be in better quality
Reviewer 3 Report
The paper presents a very interesting topic of Water- Food Nexus in India, through virtual water outflow / water footprint analysis. A quantitative (calculation of water footprint) and qualitative (analysis of publicly available policies and action plans in the public domain regarding governance of water and agriculture sectors) approach is used. The main objectives of the paper are clear and supported by data, references and explanation of the methodological steps. However, as this paper is an extension of previously published work by the same authors (e.g. 31, 39, 42), the contribution of this particular research should be emphasized more in the Introduction. Some minor issues are listed below: - The reasons for the periods selected for analysis should be explained more clearly for the reader. In Section 3, Pg. 7, L217 periods are listed as 1996-2005 and 2005-2014. It is not clear which time period 2005 belons (2006-2014?). - Tables 1-3 in section 3 with the water footprint analysis are done for the 6 spatial zones in India, but then the water and agriculture policy analysis is done at the state level. I would suggest that the analysis of water footprint should also be presented at the state level and authors should explain the rationale behind the selected three states for which the analysis has been done in section 4. - It is not clear why the analysis is done for Punjab for both periods (1996-2005, 2006-2014 in Figs. 3 and 4) and for the remaining two states only for one period. - Fig. 1 and Fig. 3-9 need better resolution for the results to be visible
I suggest that the paper should be accepted after adressing above listed issues.
Round 2
Reviewer 1 Report
Authors have done a great job to address all the comments to present this work with better clarity. I enjoyed reading this paper. I recommend for its acceptance for the publication.